# Cyclin-Dependent Kinase 5 (CDK5)-Mediated Phosphorylation of Upstream Stimulatory Factor 2 (USF2) Contributes to Carcinogenesis

**DOI:** 10.3390/cancers11040523

**Published:** 2019-04-12

**Authors:** Tabughang Franklin Chi, Tina Horbach, Claudia Götz, Thomas Kietzmann, Elitsa Y. Dimova

**Affiliations:** 1Faculty of Biochemistry and Molecular Medicine, University of Oulu, 90014 Oulu, Finland; Franklin.Tabughang@oulu.fi (T.F.C.); Thomas.Kietzmann@oulu.fi (T.K.); 2Medical Biochemistry and Molecular Biology, Saarland University, 66421 Homburg, Germany; Claudia.Goetz@uks.eu

**Keywords:** upstream stimulatory factor 2 (USF2), cyclin-dependent kinase-5 (CDK5), CRISPR-Cas9, phosphorylation, proliferation, migration

## Abstract

The transcription factor USF2 is supposed to have an important role in tumor development. However, the regulatory mechanisms contributing to the function of USF2 are largely unknown. Cyclin-dependent kinase 5 (CDK5) seems to be of importance since high levels of CDK5 were found in different cancers associated with high USF2 expression. Here, we identified USF2 as a phosphorylation target of CDK5. USF2 is phosphorylated by CDK5 at two serine residues, serine 155 and serine 222. Further, phosphorylation of USF2 at these residues was shown to stabilize the protein and to regulate cellular growth and migration. Altogether, these results delineate the importance of the CDK5-USF2 interplay in cancer cells.

## 1. Introduction

The upstream stimulatory factors (USFs), USF1 and USF2, belong to the basic helix-loop-helix leucine zipper (bHLH-Zip) family of transcription factors. They can bind as homodimers or heterodimers [1] to E-boxes with a 5’-CANNTG-3’ DNA-core sequence within the promoter of their target genes [2,3]. USFs are encoded by distinct genes [4,5,6,7] and an alternative splicing of USF2 pre-mRNA gives rise to USF2a, USF2b, and USF2c isoforms; thereby the latter two appear to act as negative regulators of gene expression [1,2,8,9,10,11]. Although USFs are expressed in virtually all tissues, their exact biological roles are still not completely understood. Both appear to play crucial roles as either transcriptional activators or transcriptional repressors of various genes [12] they can cooperate with other factors in a tissue- and/or a stimulus-specific manner (for review see [13]) to activate or inhibit the transcription of identical genes (e.g., PAI-1, HMOX-1, TERT) [14,15,16,17]. While USF1 has roles in metabolism, circadian rhythm, tanning, and the immune responses, USF2 appears to be crucial for the control of growth and developmental processes since it was shown to affect embryogenesis, fertility and growth, brain function, and iron homeostasis, [2,18,19]. Indeed, about 50% of USF2 knockout mice die shortly after birth and surviving littermates remain smaller throughout their entire lifetime than wild-type littermates [2,20] while USF1 knockout mice appear to be normal [2]. Furthermore, USF2 seems to be linked to cellular transformation and tumor progression, although a concise view has not been reached due to different results. Some studies indicated the role of USF2 as a tumor suppressor in prostate cancer [21], breast cancer [22,23] and oral cancer [24], whereas other studies suggested a role as a tumor-promoter in small cell lung cancer and squamous cell carcinoma [25], hepatocellular carcinoma (HCC) cell lines and in patients with liver cirrhosis [26] as well as in breast cancer [27]. Thus, whether USF2 acts as a tumor suppressor or tumor promoter may depend on the regulatory mechanisms involved. However, it is almost unknown which regulatory mechanisms contribute to the complex, yet not completely understood, tumor-suppressive or tumor-promoting function of USF2. Changes in the phosphorylation status are often accompanied with transcription factor regulation. Indeed, it has been shown that USF2 is a phosphoprotein [28,29] and that a number of tumor entities such as prostate cancer [30,31,32,33], hepatocellular carcinoma [34,35,36,37] and breast cancer [38,39,40] are associated with an aberrant USF2 function (for review see [13]). In addition, these cancers are also associated with high expression and activity levels of cyclin-dependent kinase-5 (CDK5) (for review see [41])). Although CDK5 is a non-typical CDK with primary functions in neurons, it has recently emerged as a key player in processes associated with carcinogenesis such as cell proliferation and metastasis. Thus, due to the correlation of USF2 and CDK5 in partially the same cancers, CDK5 seems to be of special interest in the context of USF2 regulation. However, a direct interplay between USF2 and CDK5 has not been reported. Therefore, it was the aim of the present study to determine whether USF2 can be phosphorylated by CDK5 and to investigate the impact of CDK5-dependent USF2 phosphorylation on cell proliferation and migration in human cancer cells lacking CDK5.

## 2. Results

### 2.1. USF2 is a Phosphorylation Target of CDK5

Although it is known that USF2 is a phosphoprotein and that CDK5 has an abnormal expression and activity in several human cancers, it is unknown whether USF2 is a substrate for CDK5. To examine this, we expressed full-length USF2 and several deletion variants as GST-fusion proteins in bacteria (Figure 1A) and used them as substrates in kinase assays with recombinant human CDK5 and [32P]γATP (Figure 1B). A strong phosphorylation signal was detected with full-length USF2 as well as with the USF2 protein containing only the USF-specific region (USR, 200–231); a weaker phosphorylation signal occurred with the proteins containing the N-terminal domains (1–199 and 1–161). We observed no phosphorylation with the proteins containing amino acids 157–199 or the bHLH-LZ domain (232–346). Together, these data show that USF2 can be directly phosphorylated by CDK5 and that the phosphorylated residues reside within the amino acids from 1–156 and the USR (200–231). 

To determine whether USF2 is phosphorylated by CDK5 in vivo we performed immunoprecipitation assays (IPs) in PC3 cells which were edited by the CRISPR-Cas9 technique to eliminate CDK5 (ΔCDK5). When endogenous USF2 was precipitated from the cells and probed with anti-phosphoserine and anti-phosphothreonine antibodies we found that serine- and threonine-phosphorylated USF2 was present in the WT cells (Figure 1C). Interestingly, serine-phosphorylated USF2 was nearly undetectable in ΔCDK5 cells, while threonine-phosphorylated USF2 could be detected in ΔCDK5 cells (Figure 1C). The observed effects were CDK5-specific since reintroduction of wild-type but not kinase-dead CDK5 rescued the serine phosphorylation of USF2. Thus, these data indicate that USF2 is a substrate of CDK5 in vivo and that mainly serine residues of USF2 are phosphorylated by CDK5.

### 2.2. Mapping the Exact CDK5 Phosphorylation Sites within USF2

In order to map the exact CDK5 phosphorylation sites we performed bioinformatic sequence analyses of the USF2 protein with PhosphoMotif Finder [42], and Phosida [43]. According to the kinase assays with the truncated USF2 proteins (Figure 1A,B) and the CDK5 consensus sites we found four residues (S155, T202, S222, and T230) which could represent CDK5 phosphorylation sites (Figure 2A). To identify the functionality of the predicted CDK5 phosphorylation sites within the USF2 protein, we mutated the respective serine and threonine residues separately and converted them into non-phosphorylatable alanine residues and performed kinase assays. The results from the kinase assays with these mutants and CDK5 revealed that serine 155 in the N-terminal activation domain (1–161) and serine 222 in the USR (200–231) act as CDK5 phosphorylation sites since replacement of these residues by alanine completely abolished the phosphorylation (Figure 2B). 

Since the phosphorylation sites could be important for USF2 transactivity, we performed transactivity assays with USF2 constructs in which the DNA binding domain (aa 232–346) was replaced with the Gal4 DNA binding domain and where the two CDK5 phosphorylation sites were substituted with alanine (Gal4-USF2-S155A/S222A), respectively. Only Gal4 DNA binding domain containing constructs were used as a control. The respective GAL4-USF2-WT and S155A/S222A mutant constructs were then cotransfected with a luciferase (Luc) reporter regulated by 5 Gal4 response elements (RE). Thus, changes in Luc activity reflect the changes in the transactivity of USF2. While CDK5 increased Luc activity with the USF2-WT construct, no changes could be detected with the USF2-S155A/S222A double mutant (Figure 2C). Thus, these data suggest that CDK5 can regulate USF2 transactivity.

### 2.3. The Protein Levels of USF2 are Altered in the Absence of CDK5

Next we aimed to study the functional importance of CDK5-dependent USF2 phosphorylation in the broader cellular context in the presence and absence of CDK5. 

To do this, we used, in addition to the prostate cancer cell line PC3, the hepatoma cell line HepG2 and the breast cancer cell line MDA-MB 231 in which we eliminated CDK5 by using the CRISPR-Cas9 technique (Figure 3A). Interestingly, in all these ΔCDK5 cells we detected significantly reduced USF2 protein levels compared to the corresponding scrambled control (Sc) cells (Figure 3A,B). Similarly, the reduced amount of USF2 was also visible at the nuclear level, while nuclear localization per se was not affected by the absence of CDK5 (Figure 3C). To determine whether the down-regulation of USF2 is caused by an altered transcription of the USF2 gene, USF2 mRNA was measured in all cells using quantitative RT-PCR. The mRNA levels of USF2 were not changed in ΔCDK5 PC3 and HepG2 cells in comparison with the respective control cells suggesting that lack of CDK5 does not suppress the expression of the USF2 gene. In addition, a significant increase in USF2 mRNA was observed in MDA-MB-231 ΔCDK5 cells, most probably as a compensatory mechanism for the reduced protein levels (Figure 3D). 

### 2.4. Lack of CDK5 Decreases the Half-Life of USF2

The above data suggest that CDK5-dependent USF2 phosphorylation directly affects the stability of the USF2 protein. Hence, we next investigated whether lack of CDK5 affects USF2 protein stability. To do this, we treated respective control (Sc) and ΔCDK5 PC3, ΔCDK5HepG2 and ΔCDK5 MDA-MB-231 cells with the protein synthesis inhibitor CHX and monitored the level of USF2 by western blotting. The USF2 half-life in cells lacking CDK5 was estimated to be significantly reduced (PC3ΔCDK5 t1/2 = ~8 h, HepG2ΔCDK5 t1/2 = ~11 h, MDA-MB-231ΔCDK5 t1/2 = ~12 h) when compared to the half-life of USF2 in the respective control cells (PC3 t1/2 = ~16 h, HepG2 t1/2 = ~28.5 h, MDA-MB-231 t1/2 = ~24 h) (Figure 4). These data show that the lack of CDK5 decreases the protein stability of USF2.

### 2.5. Phosphorylation of USF2 by CDK5 Affects Cell Proliferation and Migration

Next we aimed to determine what is the biological significance of CDK5-dependent USF2 phosphorylation. By measuring the proliferation rate (Figure 5A,B), BrdU incorporation (Figure 5C,D), apoptosis (Figure 5E,F) and migration (Figure 5G,H) of ΔCDK5 PC3, ΔCDK5 HepG2 and ΔCDK5 MBA-MD-231 cells in comparison to the respective control Sc cells, we found that absence of CDK5 reduced the proliferation and migration of PC3, HepG2, and MDA-MB-231 cancer cell lines in line with previous reports [33,37,38] and did not affect apoptosis (Figure 5E,F). We next established ΔCDK5 cell lines stably expressing either the non-CDK5 phosphorylatable USF2 mutant (USF2-S155A/S222A) or the phospho-mimicking USF2-S155D/S222D mutant and measured proliferation and migration. Interestingly, all cells expressing the USF2-S155A/S222A, (A/A) mutant, proliferated (Figure 5B,D), and migrated (Figure 5G,H) at a rate similar to that of control ΔCDK5 cells. Importantly, overexpression of the USF2-S155D/S222D, (D/D) mutant, rescued the lack of CDK5 and significantly increased cell proliferation (Figure 5B), BrdU incorporation (Figure 5D) and cell migration (Figure 5G,H) to a level comparable with control cells. Together, these data suggest that phosphorylation of USF2 by CDK5 can promote cancer cell proliferation and migration. 

## 3. Discussion

In this study, we explored the CDK5-mediated phosphorylation of USF2 as a regulatory mechanism, which can cause functional alterations of USF2 in cancer. We identified USF2 as a novel CDK5 target that is phosphorylated at two serine residues, Ser155 and Ser222. The CDK5-mediated USF2 phosphorylation promotes USF2 protein stability at least in three different cancer cell lines—PC3, HepG2, and MDA-MB-231. Additionally, expression of USF2 Ser155D/Ser222D, the USF2 phospho-mimicking mutant, in a CDK5 null background confers a proliferative and a migration advantage, which suggests that phosphorylation of USF2 by CDK5 can play a role during carcinogenesis. Collectively, these results establish a novel mechanism by which USF2 protein levels, and hence its function, can be regulated in cancer cells.

USF2 as a transcription factor plays important roles in processes, such as embryogenesis, metabolism and cancer development. Interestingly, USF2 has been reported to have a dual role as either tumor-suppressor or tumor-promoter. USF2 is observed in various human cancers but what mechanisms cancer cells could use to suppress or activate its function is largely unknown. Thus, understanding the mechanism(s) regulating USF2 function will help to reveal its role in the progression of cancer. In that respect, the current findings identifying USF2 as a CDK5 target provide novel insights into those mechanisms. However, CDK5 is not the only kinase having the ability to convert USF2 into a phosphoprotein and need to be seen in conjunction with earlier findings showing that also PKA [27] and GSK3β [28] can phosphorylate USF2. However, a concise view has not been reached due to different, partially cell-type specific, findings. Although none of the following studies investigated the participation and phosphorylation of USF2 directly by PKA, it appeared that cAMP and hence PKA might contribute to colorectal cancer cell proliferation [44], whereas PKA seems to be anti-tumorigenic in lung cancer cells [45]. The contribution of GSK3β and USF2 to carcinogenesis also seems to be influenced by the cellular context, although the majority of studies point to a cancer-promoting effect [25,46] where, like in the present study, GSK3β activated USF2 promoted proliferation and cell migration [29].

Interestingly, the phosphorylation signal detected was stronger with the USF2 protein containing only the USF-specific region (USR, 200–231) than with the USF protein containing the N-terminal domains (1–199 and 1–161) (Figure 1). Given that the USR is supposed to play a crucial role in transcriptional activation of USF2 [47,48,49], the stronger phosphorylation in that area reflects its importance for USF2 function. 

The analysis of the data obtained by the phosphorylation prediction software revealed four putative CDK5 phosphorylation sites within the mapped domains: Ser155 within the N-terminal activation domain and Thr201, Ser222, and Thr230 within the USF-specific region (USR). The kinase assays showed that Ser155 in the N-terminal activation domain (1–161) and Ser222 in the USR domain (200–231) are then the only CDK5 phosphorylation sites, which were also in line with the IP data using the residue-specific phospho-antibodies where only serine-phosphorylated USF2 was undetectable in ΔCDK5 cells (Figure 1)

The preferred substrate recognition motif of CDK5 was determined to be S/T-P-X-K/H/R, whereby S or T are the phosphorylatable serine or threonine residues, P is the obligatory proline residue in the +1 position, X is any amino acid and K (lysine), H (histidine), and R (arginine) are the preferred basic residues in the +3 position [50,51,52]. From the USF2 sites identified in this study, neither Ser155 (S155PAA) nor Ser222 (S222PKI) matches this theoretical CDK5 recognition motif exactly. Both sites have the required proline residue in the +1 position, but while Ser155 has no basic amino acid in its neighbourhood at all, Ser222 at least has a lysine residue in the +2 position. However, the kinase assays showed that Ser155 in the N-terminal activation domain (1–161) and Ser222 in the USR domain (200-231) are indeed the CDK5 phosphorylation sites since replacement of these residues by alanine completely abolished the phosphorylation (Figure 2). It is worth to mention that there are also other CDK5 substrates known that do not match the preferred recognition motif such as CDK5 activator p35 (having a basic amino acid in the +4 position) [53] and p53 (having no basic residue in the vicinity of the CDK5 phosphorylation site) [54]. 

Interestingly, our study also revealed that lack of CDK5 and hence lack of CDK5-mediated phosphorylation of USF2 dramatically reduces the protein levels (Figure 3) and the half-life of USF2 (Figure 4) while not inhibiting USF2 transcription. Thus, the impact of CDK5-dependent USF2 phosphorylation can also be attributed to the USF2 functions. However, the exact function of USF2 may depend on the cellular context and in particular this may be important in cancer. While few studies support the idea of USF2 being tumor-suppressive with respect to prostate [21] and breast cancer [22,23], other studies reported USF2 as being rather cancer promoting like in lung cancer [25], thyroid cancer [46], hepatocarcinoma [26]), and, surprisingly, breast cancer [27]. Though CDK5 generally contributes to the tumorigenic processes in different type of cancers (see references in [41]), an anti-proliferative function of CDK5 has been reported in human gastric cancer tissues [55]. Besides, among the substrates of CDK5 identified so far are several transcription factors such as p53, STAT3, AR, etc. (reviewed by [41]).

Our data with PC3, HepG2, and MDA-MB-231 cells with CRISPR-Cas9-mediated CDK5 knockout strongly confirmed that CDK5 affects the cellular proliferation and migration in line with previous reports using CDK5 knockdown by siRNA or inhibition by roscovitine [35,56]. The significant co-occurrence of USF2 and CDK5 (*p* < 0.001) as well as the presence of genetic alterations in both genes (3.98% amplifications and 6.25% mutations in prostate adenocarcinoma/FHCRC study, and 3.21 % of amplifications in breast cancer/Metabric study) (cbioportal.org) together with the observed enhanced proliferation and migration of all cell lines expressing the phosphomimicking USF2 Ser155D/Ser222D but not the non-phosphorylatable USF2 Ser155A/Ser222A mutant favor the tumor-promoting role of USF2. These functional data strongly imply that USF2 might be an integral component of proliferation and migration processes in prostate, hepatocellular and breast cancer cells. 

## 4. Materials and Methods

### 4.1. Materials

All biochemicals and enzymes were of analytical grade and purchased from commercial suppliers. Active recombinant GST-tagged full-length human CDK5/p35 was purchased from SignalChem (Richmond, BC, Canada). 

### 4.2. Plasmid Constructs

The generation of plasmids allowing bacterial expression of GST-tagged full-length human USF2 and USF2 deletion fusion proteins in pGEX-5X-1 (GE Healthcare, Freiburg, Germany), as well as the plasmid pcDNA6A-Gal4-USF2, have been already described [28]. The expression vectors for wild-type (Addgene #1872) and kinase-dead (Addgene #1873) CDK5 were generous gifts from Sander van den Heuvel [57]. For the generation of the USF2 point mutants, the QuikChange Site-Directed Mutagenesis Kit (Agilent Technologies, Helsinki, Finland) was used. The pLenti6/V5-DEST™ vector (Invitrogen, Karlsruhe, Germany) has been used for the generation of lentivirus particles expressing non-phosphorylatable USF2-S155A/S222A and phosphomimicking USF2-S155D/T222D proteins. 

### 4.3. Cell Culture, Transfection, and Luciferase Assays

PC3, HepG2, and MDA-MB-231 cell lines were obtained from the American Type Culture Collection (ATCC) and maintained in DMEM (Sigma-Aldrich, Helsinki, Finland), supplemented with 10% fetal bovine serum (Biochrom AG, Berlin, Germany), 1% non-essential amino acids (Sigma-Aldrich), and 0.5% ciprofloxacin (MP Biomedicals, Illkirch Cedex, France), at 37 °C in an atmosphere of 16% O2, 5% CO2, 79% N2 and 97% humidity. Transient transfections and luciferase assays were performed as described [29].

### 4.4. RNA Preparation and Quantitative Real-Time PCR Analyses

Isolation of total RNA was performed using the Qiagen RNeasy® Mini Kit (Qiagen, Hilden, Germany) following the manufacturer’s instructions. One μg of total RNA was used for cDNA synthesis with the iScript™ cDNA synthesis Kit (Bio-Rad, Helsinki, Finland). Quantitative real-time PCR was performed in an Applied Biosystems 7500 Real-Time PCR System (Life Technologies, Helsinki, Finland) by using an Applied Biosystem Power SYBR® green PCR master mix (Life Technologies). The following primers were used: USF2 forward 5′-GCGTTCGGCGACCACAATA-3′, USF2 reverse 5′-GACTACGCGGTATGTCACCT-3′, ActB forward 5′-GTTGTCGACGACGAGCG-3′, ActB reverse 5′-GCACAGAGCCTCGCCTT-3′, Hprt forward 5′-CCTGGCGTCGTGATTAGTGAT-3′, Hprt reverse 5′-AGACGTTCAGTCCTGTCCATAA-3′. All primer sets used were validated for their product and amplification efficiency using standard dilution analysis and melting curve analysis. β-Actin (ActB) and hypoxanthine-guanine phosphoribosyltransferase (Hprt) were used as internal controls to normalize the variability in expression levels. The experiments for each data point were carried out in triplicate. The relative quantification of gene expression was determined using the ΔΔCt method [58,59].

### 4.5. Generation of CDK5 Knockout PC3, HepG2, and MDA-MB-231 Cell Lines by CRISPR-Cas9-Mediated Genome Editing

A 20-bp guide sequence targeting the second exon of human CDK5 (CDK5-201, ENST00000297518.4 and CDK5-204, ENST00000485972.5) with the sequence 5′-CCGGGAGACTCATGAGATCG-3′ was designed online using Zhang’s laboratory web resource (www.genome-engineering.org); a non-targeting, scrambled sequence 5′-GCACTACCAGAGCTAACTCA-3′ (OriGene) was used as a negative control. gRNA-encoding oligonucleotides (Sigma-Aldrich) were cloned into the vector SpCas9(BB)-2A-GFP (PX458, Addgene plasmid ID 48138) using standard procedures as described [60]. The generation of the CDK5 control and knockout cells via CRISPR-Cas9-mediated non-homologous end-joining (NHEJ) DNA repair and the screening was performed according to described guidelines [60]. In brief, the PC3, HepG2, and MDA-MB-231 cells were transiently transfected with FuGENE HD transfection reagent (Promega, Finland) and either the genome editing or the scrambled CRISPR-Cas9 construct according to the manufacturer’s instructions. 48 h post-transfection cells were subjected to single-cell-sorting by using fluorescence-based flow cytometry (BD FACSAria™ III cell sorter). The single-cell clones were expanded and screened for frame-shift mutations; shortly, a region spanning the target site was amplified by PCR from genomic DNA isolated from clonal cell lines using the following forward 5′- CCTGCCACATCCTCACTCAC-3′ and reverse 5′-AGGGCAAAAGAAGGGCTCAG-3′ primers. PCR products were subsequently cloned into pUC19 (Invitrogen). 15–20 sequences were analyzed per clone by aligning them to the WT CDK5 sequences using BLAST and Serial Cloner. The sequences of primers used for screening of the off-targets are: off-target 1 (F1 5′-CCTCCCTTCTCACCTCTGGA-3′, R1 5′-CAGGAAGCTCCTCCTACCCT-3′), off-target 2 (F2 5′-TGCAAGGATTCAAGCTCCGA-3′, R2 5′-CTGCCCCCGATTATAGAGGC-3′), off-target 3 (F3 5′-CCCCTGAGCCTTATCGCAAA-3′, R3 5′-CACTGTGCCAGGATCTGTGA-3′), and off-target 4 (F4 5′-GGAAGGCACCTATGGGACAG-3′, R4 5′-AGTGTGACCACTCCAGCTCA-3′).

### 4.6. Protein Isolation, Immunoprecipitations, Half-Life Studies, Immunofluorescence, and Western Blots

Cells were washed with ice cold 1x PBS, scraped in lysis buffer (50 mM Tris pH 8.0, 5 mM EDTA, 150 mM NaCl, 5 mM DTT, complete mini protease inhibitors (Roche, Helsinki, Finland), incubated for 20 min on a lab-roller at 4 °C and then centrifuged at 13,000× *g* for 20 min. The Bradford method was used for measuring the protein concentration.

In half-life studies, 10 µg/mL cycloheximide (CHX, Sigma Aldrich, Helsinki, Finland) was added to the cell culture medium of WT and ΔCDK5 cells for the indicated periods. After harvesting the cells, endogenous USF2 protein levels were measured by western blot analyses and quantified with Fiji (NCBI) software.

For immunohistochemistry, cells were placed on coverslips and fixed by ice-cold methanol/acetone (1:1) for 10 min, blocked with 3% BSA in PBS, and incubated with an anti-USF2 mAb (N-18; 1:500; Santa Cruz Biotechnology, Heidelberg, Germany) at room temperature for 1 hr. Subsequently, cells were washed in PBS three times for 10 min before incubation with FITC-conjugated goat anti-rabbit IgG secondary Ab (1:400; #A16143 Thermo-Fisher). Thereafter, cells were washed three times again with PBS for 10 min. DAPI (1 µg/mL) was added in the final wash. 

For immunoprecipitation experiments, protein samples from WT and ΔCDK5 PC3 cells were prepared as already described [28] and subjected to western blot analyses. 

For the analysis of proteins by western blotting, cell extracts were denaturated with a sample buffer (500 mM Tris pH 6.8, 30% glycerol, 10% SDS, 0.01% bromophenol blue, 40 mM DTT) by incubating at 95 °C for 10 min. Then, the proteins were resolved at SDS-PAGE and blotted to a nitrocellulose membrane. As primary antibodies polyclonal antibodies against CDK5 (C-8; 1:1000; Santa Cruz Biotechnology), USF-2 (N-18; 1:500; Santa Cruz Biotechnology), phospho-serine (#61-8100; 1:200; Invitrogen), HA-tag (1:500, Santa Cruz Biotechnology) as well as monoclonal antibodies against phospho-threonine (#P6623; 1:300; Sigma Aldrich), V5 tag (#37-7500, 1:5000; Thermo-Fischer, Helsinki, Finland), and α-tubulin (B-5-1-2; 1:10000; Sigma-Aldrich) were used. Horseradish peroxidase (HRP)-conjugated goat anti-rabbit and goat anti-mouse IgGs (both 1:5000; Bio-Rad) were used. The enhanced chemiluminescence (ECL) system (GE Healthcare) was used for detection.

### 4.7. Expression and Purification of Proteins, Radioactive Kinase Assays

GST-tagged full-length human USF2, USF2 deletion, and USF2 double point mutation fusion proteins were expressed in *E.coli* BL21 (DE3) at 30 °C and purified with glutathione sepharose beads as described previously [28]. Radioactive kinase assays were carried out as described [29]. In brief, purified GST fusion proteins (1.5 µg) were incubated in kinase assay buffer (15 mM MOPS pH 7.2, 15 mM MgCl2, 3 mM EGTA, 1.2 mM EDTA, 150 µM DTT, 6 mM β-glycerophosphate) containing 20 ng of active, recombinant full-length human CDK5/p35 (SignalChem), 50 µM unlabeled ATP and 1.5 µCi (γ-32P) ATP at 30 °C for 20 min. The proteins were denaturated by incubating with an SDS-containing sample buffer at 95 °C for 10 min. Following separation of the proteins by 10% SDS-PAGE, the gel was exposed to a Kodak phosphor imaging screen overnight and thereafter autoradiography was detected with a Molecular Imager FX using the Quantity One software (all BIO-RAD). Afterward, the total proteins were detected by Coomassie staining.

### 4.8. Overexpression of USF2-S155A/S222A and USF2-S155D/S222D Double Point Mutants in Cells via Lentiviral Infection

24 h post-transfection, the lentivirus containing medium was collected and filtered. Prior infection of cells using a single infection with 6 mL virus, polybrene was added (1 µg / mL). Infected cells were subjected to selection with 0.5 μg/mL blasticidin for 2–3 weeks before harvesting and analyses. The overexpression of USF2 was verified by western blot analysis. 

### 4.9. Proliferation and Cell Migration Assay

Proliferation assays were performed by counting viable cells using a hemocytometer as well as by measuring BrdU incorporation. For counting, cells were seeded in triplicate in 6-well plates at a density of 30,000 cells/well. Cells were trypsinized and counted each day for 4 days at the same chronological time (as indicated in the results section). The BrdU assay (Calbiochem, Helsinki, Finland) was performed according to the manufacturer’s instructions and the absorbance was measured at dual wavelengths of 450–540 nm.

Cell migration was measured by using medium-treated 24-well Transwell chambers (BD Bioscience, Heidelberg, Germany) with 8.0 µm polycarbonate membranes. The bottom chamber was filled with 600 µL medium containing 10% fetal calf serum; cells were seeded into the top chamber at a density of 1 × 104 cells per well in 100 µL serum-free medium. After incubation in a humidified incubator with 5% CO2 at 37 °C for 4 h, cells were fixed with 4% paraformaldehyde and the non-migratory cells were scraped off from the top of the Transwell with a cotton swab. The migrated cells attached to the bottom chamber were stained with 0.0075% crystal violet. Crystal violet absorption at 595 nm was measured in a Tecan Infinite® M1000Pro microplate reader. For each migratory condition, two identical replicates were performed.

### 4.10. Apoptosis Analyses

Cell apoptosis was measured by Annexin-V-FLOUS staining kit (Roche/Sigma-Aldrich, Helsinki, Finland) according to the manufacturer‘s protocol. Apoptotic cells were analyzed by flow cytometry (BD FACSAria™ III cell sorter).

### 4.11. Statistical Analysis

Each experiment was performed at least three times and representative data are shown. Data in bar graphs are given as mean values ± standard deviation (SD). Statistical differences were calculated by using the Student t-test with error probabilities of *p* < 0.05 to be significant.

## 5. Conclusions

The findings described in this study demonstrate the important role for CDK5-mediated phosphorylation of USF2 and provided a mechanism by which USF2 protein levels can be regulated in cells.

## Figures and Tables

**Figure 1 cancers-11-00523-f001:**
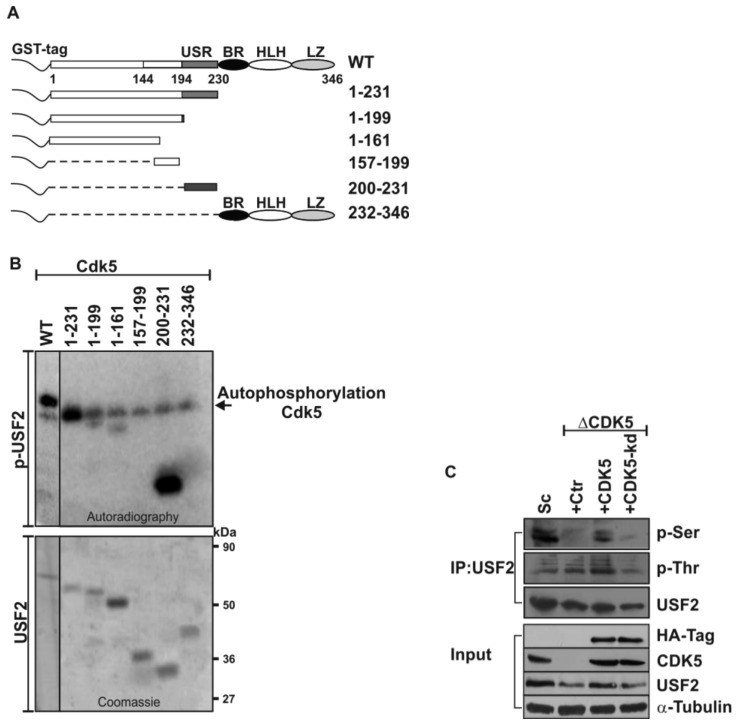
CDK5 phosphorylates USF2. (**A**) Scheme of USF2, its domains and deletion mutants that were cloned, expressed, and purified as GST fusion proteins. (**B**) Purified GST-tagged full-length USF2 and its deletion mutants were phosphorylated with recombinant human CDK5 (upper panel) in the presence of [32P]-γ-ATP for 20 min at 30 °C. Proteins were separated by 10% SDS-PAGE and radioactivity was detected by autoradiography. The bands corresponding to the phosphorylated proteins (1–231) and (1–199) were partially covered by auto-phosphorylated Cdk5. Attempts to separate the bands by using 7.5% or 12% SDS-PAA gels were not successful. The lower panel shows the Coomassie staining performed as a loading control. (**C**) Endogenous USF2 was immunoprecipitated from control (Sc), ΔCDK5 PC3 cells and ΔCDK5 cells transfected with a plasmid encoding wild-type or kinase-dead CDK5; phospho-USF2 protein levels were detected with phospho-threonine or phospho-serine antibodies.

**Figure 2 cancers-11-00523-f002:**
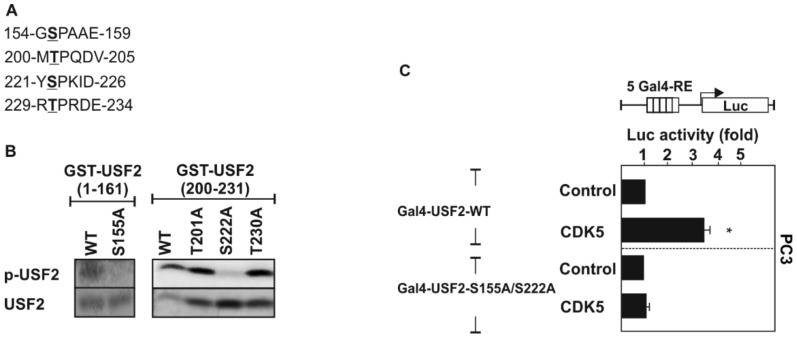
USF2 is phosphorylated at Ser155 and Ser222 by CDK5. (**A**) In silico predicted CDK5 phosphorylation sites in human USF2 (**B**) Purified GST-tagged USF2 variants with point mutations at the predicted CDK5 phosphorylation sites were phosphorylated with recombinant human CDK5 in the presence of [32P]-γ-ATP for 20 min at 30 °C. Proteins were separated by 10% SDS-PAGE and radioactivity was detected by autoradiography. The lower panel shows the recombinant USF2 proteins on a Coomassie-stained gel performed as a loading control. (**C**) PC3 cells were cotransfected with pFR-5Gal4-RE-Luc, an expression vector for CDK5 and constructs expressing WT or mutant pcDNA6-Gal4-USF2 (1–231) or the appropriate empty Gal4 vector. The luciferase activity was calculated as fold induction compared to the Gal4-USF2 (1–231)-WT luciferase activity after subtracting the values from the empty Gal4 expression vector. *, significant differences control vs. CDK5.

**Figure 3 cancers-11-00523-f003:**
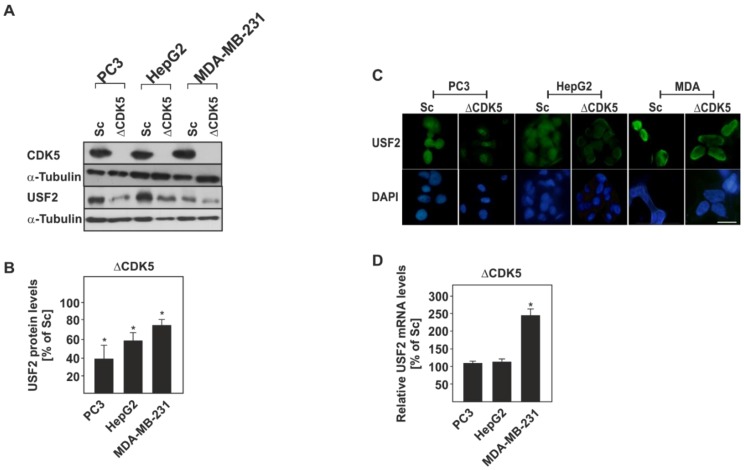
Lack of CDK5 alters USF2 protein and mRNA levels in PC3, HepG2, and MDA-MB-231 cancer cell lines. (**A**) Representative western blot of USF2 protein levels. (**B**) Quantification of USF2 protein levels. USF2 levels in all Sc cells were set to 100%. Values represent means ± SD of at least three independent experiments. (**C**) Representative fluorescence images from the respective Sc and ΔCDK5 PC3, HepG2 and MDA-MB-231 cancer cell lines probed with an antibody against USF2 and stained with DAPI. Bar, 10 µm. (**D**) USF2 mRNA levels were measured by quantitative RT-PCR and mRNA levels in all Sc cells were set to 100%. Values represent means ± SD of at least three independent experiments. Statistics: Student’s t-test for paired values; *, significant difference Sc versus CDK5 knockout cells; *p* ≤ 0.05.

**Figure 4 cancers-11-00523-f004:**
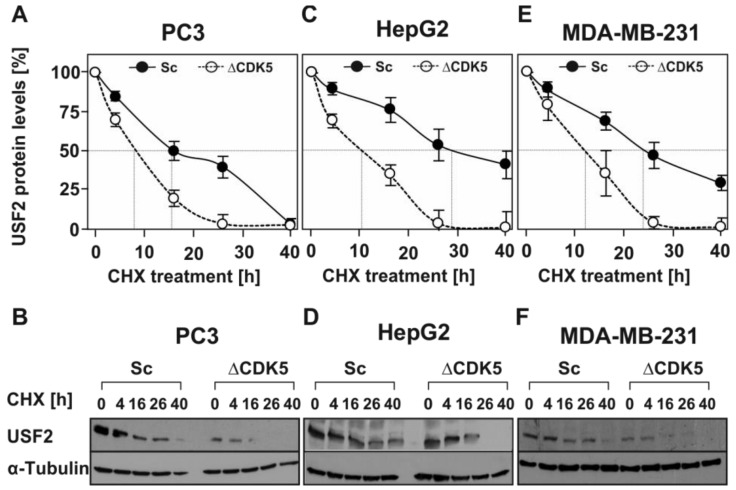
Absence of CDK5 reduces the half-life of USF2. (**A**,**C**,**E**) Respective Sc and ΔCDK5 cells were treated with 10 µg/mL CHX for the indicated time periods. Proteins were isolated, separated by SDS-PAGE and detected by western blotting. Protein levels were quantified and the relative protein level of USF2 was blotted against the duration of CHX treatment for estimation of the half-life. The dashed line indicates the USF2 half-life where 50% of the USF2 protein level was reached. (**B**,**D**,**F**) Representative western blots. 50 µg of protein was probed with an antibody against USF2 and α-tubulin.

**Figure 5 cancers-11-00523-f005:**
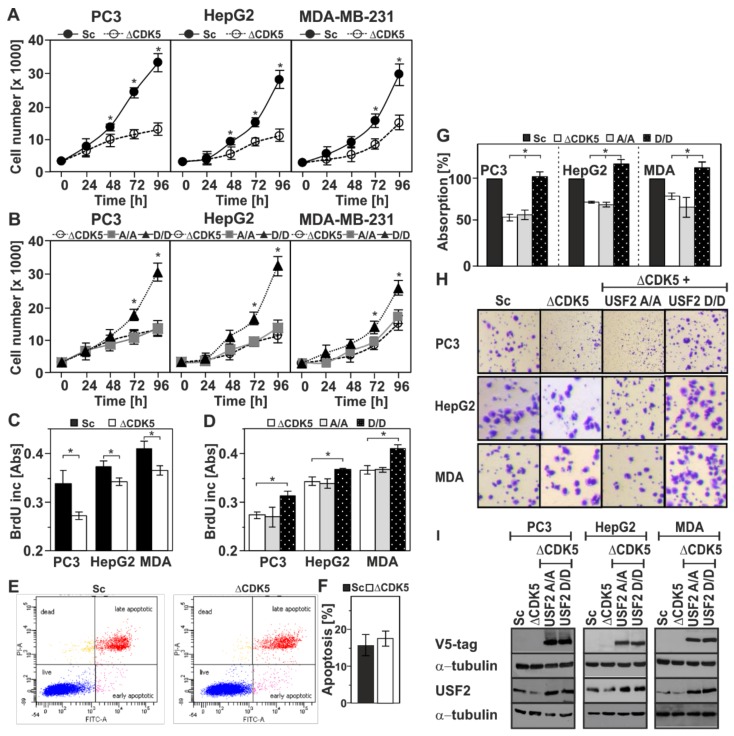
Phosphorylation of USF2 by CDK5 affects cell proliferation and migration. Cellular proliferation was monitored by cell number (**A**,**B**) and BrdU incorporation (**C**,**D**) in Sc and ΔCDK5 cells (**A**,**C**) and in ΔCDK5 cells expressing either non-phosphorylatable USF2-S155A/S222A or phosphomimetic USF2-S155D/S222D double mutants (**B**,**D**). (**E**,**F**) Apoptosis in Sc and ΔCDK5 PC3 cells was assessed by Anexin-V/PI staining and analyzed by flow cytometry; (**G**) Migration was assessed by crystal violet staining. The values represent the absorbance of crystal violet at 595 nm. *, significant difference, *p* ≤ 0.05. (**H**) Representative images of a Transwell migration assay. (**I**) The expression of USF2 was controlled by western blot with an antibody against V5-tag and USF2; α-tubulin served as a loading control.

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
