# Peer review of "Cyclin-Dependent Kinase 5 (CDK5)-Mediated Phosphorylation of Upstream Stimulatory Factor 2 (USF2) Contributes to Carcinogenesis"

_cancers, 2019, doi:10.3390/cancers11040523_

Round 1
Reviewer 1 Report
This manuscript mainly presents a novel CDK5-USF2 signaling that contributes to carcinogenesis. These results would provide clue to producing potent anti-cancer drugs. Therefore, the manuscript should be accepted as it is.
Author Response
Reviewer 1
This manuscript mainly presents a novel CDK5-USF2 signaling that contributes to carcinogenesis. These results would provide clue to producing potent anti-cancer drugs. Therefore, the manuscript should be accepted as it is.
Response: We appreciate the very positive comments of this reviewer.
Reviewer 2 Report
In the article by Franklin Chi et al. the authors provide novel insights into USF2 regulation as well as a potential mechanism on how this transcription factor could contribute to tumorigenesis in several cancer cell line models. Concretely, the authors show that USF2 is a CDK5 substrate, that CDK5 phosphorylates USF2 at two serine residues within the USR domain, and that phosphorylation of USF2 by CDK5 promotes USF2 protein stability. Lastly, in a subset of rescue experiments, the authors show that expression of a USF2 double phosphomimetic mutant rescues cell proliferation and migration in CDK5 knockout (KO) cells.
This is a well-written, easy to follow study, suitable for publication in Cancers. However, some issues arose during the revision of this manuscript which are detailed below.
Main comments
Figure 1C – How many independent CDK5-KO clones have the authors tried? Have the authors tried CDK5 inhibitors to show reduction in USF2 phosphorylation? In addition, rescue of USF2 phosphorylation in CDK5-KO cells by re-introduction of CDK5-WT, but not of a kinase-dead mutant would be a nice control to include.
Figure 2B – It is not clear that CDK5 phosphorylates USF2 Ser155 and Ser222 to the same extent. In addition, it can be inferred from figure 1B that the construct GST-USF2 (1-161) used to verify phosphorylation at Ser155 has a very similar electrophoretic mobility than recombinant CDK5 (see arrow indicating CDK5 autophosphorylation), which makes it difficult to interpret the left autoradiograph. To convincingly validate that Ser155 and Ser222 are the only two sites that CDK5 phosphorylates on USF2, the authors should conduct the in vitro kinase assay in the presence of full-length GST-USF2 WT, and the single and double S to A mutants (i.e. S155A, S222A or S155A;S222A) as substrates. Levels of recombinant CDK5 should be shown.
Figures 3 and 4 – It is convincingly shown that lack of CDK5 reduces USF2 protein stability. Have the authors tried to rescue this effect by re-expressing WT or kinase-dead versions of CDK5? Are USF2 protein levels in CDK5-KO cells rescued by treatment with a proteasome inhibitor? Does treatment with CDK5 inhibitors have the same effect on USF2 protein levels? Lastly, have the authors tried if phosphorylation of USF2 by CDK5 affects its nuclear translocation and/or transcriptional activity?
Figure 5B/C – Have the authors evaluated the effect of CDK5 depletion and re-expression of USF2 A/A or D/D mutants on cell death, e.g. apoptosis? Does expression of any single phosphomimetic mutant (i.e. S155D or S222D) rescue phenotypic effects of CDK5 KO?
Figure 5E – The authors should provide quantification (i.e. graph) of the independent migration experiments; from the image presented in panel E, it seems that KO of CDK5 does not significantly affect migration of HepG2 nor MDA-MB-231 cells.
Figure 5F – could the authors clarify in which cell line they have assessed over-expression of USF2 A/A and D/D mutants by western blot? To which extent is USF2 overexpressed in these cells compared to endogenous levels of Sc controls? This is an important control for the interpretation of these experiments, and to exclude that the rescued phenotypes are the result of USF2 expression far above of physiologically relevant levels.
Additional comments
Introduction – The authors could briefly expand on whether the different USF2 isoforms have distinct functionalities. In addition, the authors should provide additional insight into the rationale that led them to investigate USF2 as a potential CDK5 substrate.
Page 2, lines 76-80: the authors need to correct the text to clarify that there are no changes in USF2 threonine phosphorylation following CDK5 KO, so that the text is consistent with figure 1C findings.
Figure 5F – Western blots showing expression of USF2 A/A and D/D mutants in CDK5 KO cells should be shown for all cell lines used across figure 5.
Material and methods – Section 4.6: source of V5 antibody is not stated in the text. Section 4.8: the authors should clarify whether the D/D mutant involves T230 or S222.
Discussion – the authors could briefly explain how phosphorylation by other protein kinases (PKA, GSK3B) affects USF2 activity or contribution to tumorigenesis, if this is known.
Discussion – Could the authors show the data from cBioPortal regarding the co-occurrence of CDK5 and USF2? At which level do they correlate (copy number, mRNA expression…)?
Author Response
Reviewer 2
Main comments
Figure 1C – How many independent CDK5-KO clones have the authors tried? Have the authors tried CDK5 inhibitors to show reduction in USF2 phosphorylation? In addition, rescue of USF2 phosphorylation in CDK5-KO cells by re-introduction of CDK5-WT, but not of a kinase-dead mutant would be a nice control to include.
Response:
We have analyzed at least two independent cell clones each; however, to avoid too much redundancy we have presented the data from only one clone each of the different cell types. With respect to the use of CDK5 inhibitors, we have used roscovitine, which is widely used to inhibit CDK5 but which has been reported also to inhibit other cyclin-dependent kinases. We have observed similar effects on USF2 but did not include these data into the current manuscript due to the less-specific action of roscovitine compared to absence of CDK5. The reviewer proposed to rescue USF2 phosphorylation by re-introduction of WT-CDK5 vs a kinase-dead CDK5. This is a nice proposal; however, we think that the reviewer agrees that those experiments cannot be done properly in the 10 days’ time frame given to revise this article.
Figure 2B – It is not clear that CDK5 phosphorylates USF2 Ser155 and Ser222 to the same extent. In addition, it can be inferred from figure 1B that the construct GST-USF2 (1-161) used to verify phosphorylation at Ser155 has a very similar electrophoretic mobility than recombinant CDK5 (see arrow indicating CDK5 autophosphorylation), which makes it difficult to interpret the left autoradiograph. To convincingly validate that Ser155 and Ser222 are the only two sites that CDK5 phosphorylates on USF2, the authors should conduct the in vitro kinase assay in the presence of full-length GST-USF2 WT, and the single and double S to A mutants (i.e. S155A, S222A or S155A;S222A) as substrates. Levels of recombinant CDK5 should be shown.
Response:
We apologize, but there may have been some misunderstanding or an unintended change in the figures arrow location during file conversion. In fact, it is WT-USF2 that has a similar electrophoretic mobility than CDK5. The GST-USF2 (1-161) is certainly migrating faster than CDK5 (see clearly distinguishable two bands in lane 4 of Fig 1B). Hence, by using overlapping constructs for USF2 as displayed in Fig.1 we located the areas in which CDK5 phosphorylates USF2. The data in Fig.2B are then specifically addressing which residues in the USF2 fragments are phosphorylated; due to the size difference between GST-USF2 (1-161) and CDK5 we could clearly distinguish between autophosphorylated CDK5 and phosphorylated GST-USF2 (1-161) and also see the loss of phosphorylation upon mutation at Ser155. We think that the reviewer agrees that cropping the autoradiograph to display the difference between the respective wild-type and mutant GST-USF2 (1-161) is appropriate and apologize again for the misunderstanding caused by the misplaced arrow.
Figures 3 and 4 – It is convincingly shown that lack of CDK5 reduces USF2 protein stability. Have the authors tried to rescue this effect by re-expressing WT or kinase-dead versions of CDK5? Are USF2 protein levels in CDK5-KO cells rescued by treatment with a proteasome inhibitor? Does treatment with CDK5 inhibitors have the same effect on USF2 protein levels? Lastly, have the authors tried if phosphorylation of USF2 by CDK5 affects its nuclear translocation and/or transcriptional activity?
Response:
The reviewer raises interesting points. However, we have not tried to rescue these effects by the respective CDK5 variants; however, we know that the use of the proteasome inhibitor MG132 partially rescues USF2 protein levels. For the reasons indicated above, we have not used roscovitine or other CDK5 inhibitors in experiments investigating the USF2 protein half-life. In line with the proposal of the reviewer we have been looking into the nuclear translocation and transcriptional activity of USF2. While nuclear translocation was not affected, we could see a reduction in USF2 transactivity.
Figure 5B/C – Have the authors evaluated the effect of CDK5 depletion and re-expression of USF2 A/A or D/D mutants on cell death, e.g. apoptosis? Does expression of any single phosphomimetic mutant (i.e. S155D or S222D) rescue phenotypic effects of CDK5 KO?
Response:
The reviewer raised an interesting point. Indeed, we have looked at apoptotic parameters and it appears that there are no effects on it. Thus, the effects of CDK5 knockout affect mainly the proliferation and migration as shown in this manuscript.
Figure 5E – The authors should provide quantification (i.e. graph) of the independent migration experiments; from the image presented in panel E, it seems that KO of CDK5 does not significantly affect migration of HepG2 nor MDA-MB-231 cells.
Response:
We appreciate this suggestion and provide now quantifications of the migration experiments. They can be found in the revised figure 5.
Figure 5F – could the authors clarify in which cell line they have assessed over-expression of USF2 A/A and D/D mutants by western blot? To which extent is USF2 overexpressed in these cells compared to endogenous levels of Sc controls? This is an important control for the interpretation of these experiments, and to exclude that the rescued phenotypes are the result of USF2 expression far above of physiologically relevant levels.
Response:
We agree and provide now the expression analyses for all cell lines studied along with endogenous USF2 levels.
Additional comments
Introduction – The authors could briefly expand on whether the different USF2 isoforms have distinct functionalities. In addition, the authors should provide additional insight into the rationale that led them to investigate USF2 as a potential CDK5 substrate.
Response:
We are grateful for this suggestion and added respective parts to the introduction of the revised paper (l 31-32). In addition, we added more to the rationale why we investigated USF2 as a CDK5 substrate (l, 52-59).
Page 2, lines 76-80: the authors need to correct the text to clarify that there are no changes in USF2 threonine phosphorylation following CDK5 KO, so that the text is consistent with figure 1C findings.
Response: That has been corrected.
Figure 5F – Western blots showing expression of USF2 A/A and D/D mutants in CDK5 KO cells should be shown for all cell lines used across figure 5.
Response:
As explained above, we provide now the expression analyses for all cell lines studied along with endogenous USF2 levels.
Material and methods – Section 4.6: source of V5 antibody is not stated in the text. Section 4.8: the authors should clarify whether the D/D mutant involves T230 or S222.
Response:
We provide now the source of the V5-tag antibody. In addition, we apologize for the error in section 4.8 and indicate that the D/D mutant is a USF2 mutant where the two CDK5 sites S155 and S222 were mutated.
Discussion – the authors could briefly explain how phosphorylation by other protein kinases (PKA, GSK3B) affects USF2 activity or contribution to tumorigenesis, if this is known.
Response:
We have now added a short part indicating how phosphorylation by PKA and GSK3B affects USF2 activity and tumorigenesis (l205-212).
Discussion – Could the authors show the data from cBioPortal regarding the co-occurrence of CDK5 and USF2? At which level do they correlate (copy number, mRNA expression…)?
Response:
We have now indicted the genetic alterations of 3.98% amplifications and 6.25% mutations in the prostate adenocarcinoma/FHCRC study and 3.21% of amplifications in the breast cancer/Metabric study.
Reviewer 3 Report
Chi et.al. report that USF2 is a substrate of CDK5 which phosphorylates USF2 at two ser residues. The authors provide evidence that this CDK5 mediated phosphorylation is important for USF2 stability post-translation leading to faster cell proliferation and invasion. Overall a good manuscript although a few issues need to be addressed:
1. How does CDK5 mediated phosphorylation affect USF2 activity? Recently Tan et.al. (PMID:30244169) (Ref26) reported that USF2 promotes breast carcinogenesis by downregulating Smurf1 and Smurf2 thereby modulating TGF-β and BMP signaling. Do the authors see alterations in Smurf2 protein levels upon expression of USF2 D/D and not USF2 A/A? This would provide more mechanistic evidence of how CDK5-USF2 promotes carcinogenesis. This experiment is a quick, easy, and strongly recommended.
2. The loss of pSer on USF2 in delCDK5 is very dramatic. Please repeat Fig 1C by reintroducing CDK5 back into the delCDK5 to show that the pSer is rescued. This will help rule out Crispr guide off-targets.
3. Showing a single representative field of cells in Fig 5E doesnot provide the full information. Please quantify the migration assay over various fields…..this will help derive statistical significance.
4. The authors are strongly recommended to include all Crispr/Cas9 guide sequences and surveyor primers used in the Materials sections of the manuscript.
Author Response
Reviewer 3
Chi et.al. report that USF2 is a substrate of CDK5 which phosphorylates USF2 at two ser residues. The authors provide evidence that this CDK5 mediated phosphorylation is important for USF2 stability post-translation leading to faster cell proliferation and invasion. Overall a good manuscript although a few issues need to be addressed:
Response:
We appreciate the positive comments of the reviewer.
1. How does CDK5 mediated phosphorylation affect USF2 activity? Recently Tan et.al. (PMID:30244169) (Ref26) reported that USF2 promotes breast carcinogenesis by downregulating Smurf1 and Smurf2 thereby modulating TGF-β and BMP signaling. Do the authors see alterations in Smurf2 protein levels upon expression of USF2 D/D and not USF2 A/A? This would provide more mechanistic evidence of how CDK5-USF2 promotes carcinogenesis. This experiment is a quick, easy, and strongly recommended.
Response:
We appreciate the comment of the reviewer to look for SMURF1 and SMURF2 expression upon expression of USF2D/D and USF2A/A in CDK5-KO cells. This is a nice proposal; however, we think that the reviewer agrees that those experiments cannot be done properly in the 10 days’ time frame given to revise this article.
2. The loss of pSer on USF2 in delCDK5 is very dramatic. Please repeat Fig 1C by reintroducing CDK5 back into the delCDK5 to show that the pSer is rescued. This will help rule out Crispr guide off-targets.
Response:
We appreciate this proposal of the reviewer, however, due to the time constrains, 10 days for resubmission, we could not repeat this experiment.
3. Showing a single representative field of cells in Fig 5E does not provide the full information. Please quantify the migration assay over various fields, this will help derive statistical significance.
Response:
We appreciate this suggestion and provide now quantifications of the migration experiments. They can be found in the revised figure 5
4. The authors are strongly recommended to include all Crispr/Cas9 guide sequences and surveyor primers used in the Materials sections of the manuscript.
Response:
We appreciate this recommendation and included the respective guide sequences and primers into the Materials and Methods section.
Round 2
Reviewer 2 Report
I appreciate the authors’ effort to address most of my concerns and consider that their manuscript has significantly improved from its previous version. However, there are still some minor issues that should be fixed before acceptance for publication.
Comments
1. Figure 1C. I agree with the authors on that rescue experiments are challenging. However, to exclude any off-target effects of their CRISPR approach, the re-expression of CDK5-WT and kinase dead in their CDK5-KO cells to analyze USF2 phosphorylation would be a very important control to include.
2. Figures 3-4: the authors claim that CDK5-KO did not affect USF2 nuclear translocation, while it reduced its transactivation. These are important conclusions that should be included in the manuscript to show the whole effect of CDK5 depletion on USF2 regulation.
3. Figure 5: the authors should include their experiments showing that apoptosis is not affected. While this might be seen as a “negative result” it would help the reader to thoroughly understand the phenotypes that CDK5 regulates via USF2.
4. I appreciate the authors have included information from cBioPortal in their discussion section (lines 240-242); however, the authors should clarify if the indicated percentages are attributed to USF2 or CDK5 or both.
Author Response
We thank you for the careful evaluation of our revised manuscript “Cyclin-dependent kinase 5 (CDK5)-mediated phosphorylation of upstream stimulatory factor 2 (USF2) contributes to carcinogenesis”. Comments from the reviewer: 1. Figure 1C. I agree with the authors on that rescue experiments are challenging. However, to exclude any off-target effects of their CRISPR approach, the re-expression of CDK5-WT and kinase dead in their CDK5-KO cells to analyze USF2 phosphorylation would be a very important control to include. Response: We agree with this proposal, we apologize and we think that the reviewer agrees that those experiments cannot be done properly in the 5 days’ time frame given by the Editor to revise this article. 2. Figures 3-4: the authors claim that CDK5-KO did not affect USF2 nuclear translocation, while it reduced its transactivation. These are important conclusions that should be included in the manuscript to show the whole effect of CDK5 depletion on USF2 regulation. Response: As suggested by the reviewer, we have now incorporated the data about the transactivation (Fig.2 C) and the nuclear translocation (Fig. 3C) into the revised manuscript. 3. Figure 5: the authors should include their experiments showing that apoptosis is not affected. While this might be seen as a “negative result” it would help the reader to thoroughly understand the phenotypes that CDK5 regulates via USF2. Response: We appreciate this point and we have included the apoptosis data as part E, and F into the revised Fig. 5. 4. I appreciate the authors have included information from cBioPortal in their discussion section (lines 240-242); however, the authors should clarify if the indicated percentages are attributed to USF2 or CDK5 or both. Response: Thank you for pointing this out, we have now indicated that the changes apply to both genes.